# Monte-Carlo Planning and Learning with Language Action Value Estimates

**Youngsoo Jang**[1]**, Seokin Seo**[2]**, Jongmin Lee**[1]**, Kee-Eung Kim**[1,2]
[1]School of Computing, KAIST, Daejeon, Republic of Korea
[2]Graduate School of AI, KAIST, Daejeon, Republic of Korea
{ysjang,siseo,jmlee}@ai.kaist.ac.kr, kekim@kaist.ac.kr

## Abstract

Interactive Fiction (IF) games provide a useful testbed for language-based reinforcement learning agents, posing significant challenges of natural language understanding, commonsense reasoning, and non-myopic planning in the combinatorial search space. Agents using standard planning algorithms struggle to play IF games due to the massive search space of language actions. Thus, language-grounded planning is a key ability of such agents, since inferring the consequence of language action based on semantic understanding can drastically improve search. In this paper, we introduce Monte-Carlo planning with Language Action Value Estimates (MC-LAVE) that combines Monte-Carlo tree search with language-driven exploration. MC-LAVE concentrates search effort on semantically promising language actions using locally optimistic language value estimates, yielding a significant reduction in the effective search space of language actions. We then present a reinforcement learning approach built on MC-LAVE, which alternates between MC-LAVE planning and supervised learning of the self-generated language actions. In the experiments, we demonstrate that our method achieves new high scores in various IF games.

## 1 Introduction

Building an intelligent goal-oriented agent that can perceive and react via natural language is one of the grand challenges of artificial intelligence. In pursuit of this goal, we consider Interactive Fiction (IF) games (Nelson, 2001; Montfort, 2005), which are text-based simulation environments where the agent interacts with the environment only through natural language. They serve as a useful testbed for developing language-based goal-oriented agents, posing important challenges such as natural language understanding, commonsense reasoning, and non-myopic planning in the combinatorial search space of language actions. IF games naturally have a large branching factor, with at least hundreds of natural language actions that can affect the simulation of game states. This renders naive exhaustive search infeasible and raises the strong need for *language-grounded* planning ability, i.e. effective search space is too large to choose an optimal action, without inferring the future impact of language actions by understanding the environment state described in natural language.

Still, standard planning methods such as Monte-Carlo tree search (MCTS) are language-agnostic and rely only on uncertainty-driven exploration, encouraging more search on less-visited states and actions. This simple uncertainty-based strategy is not sufficient to find an optimal language action under limited search time, especially when each language action is treated as a discrete token. On the other hand, recent reinforcement learning agents for IF games have started to leverage pre-trained word embeddings for language understanding (He et al., 2016; Fulda et al., 2017; Hausknecht et al., 2020) or knowledge graphs for commonsense reasoning (Ammanabrolu & Hausknecht, 2020), but their exploration strategies are still limited to the $\epsilon$-greedy or the softmax policies, lacking more structured and non-myopic planning ability. As a consequence, current state-of-the-art agents for IF games still have not yet been up to the human-level play.

In this paper, we introduce Monte-Carlo planning with Language Action Value Estimates (MC-LAVE), a planning algorithm for the environments with text-based interactions. MC-LAVE combines Monte-Carlo tree search with *language-driven* exploration, addressing the search inefficiency

attributed to the lack of language understanding. It starts with credit assignment to language actions via Q-learning of the experiences collected from the past searches. Then, MC-LAVE assigns *non-uniform* search priorities to each language action based on the optimistically aggregated Q-estimates of the past actions that share similar meanings with the candidate action, so as to focus more on the semantically promising actions. This is in contrast to the previous methods that involve language understanding in the form of a knowledge graph, where the insignificant language actions are *uniformly* filtered out by the graph mask (Ammanabrolu & Hausknecht, 2020; Ammanabrolu et al., 2020). We show that the non-uniform search empowered by language understanding in MC-LAVE yields better search efficiency while not hurting the asymptotic guarantee of MCTS.

We then present our reinforcement learning approach that uses MC-LAVE as a strong policy improvement operator. Since MCTS explores the combinatorial space of action sequences, its search results can be far better than the simple greedy improvement, as demonstrated in the game of Go (Silver et al., 2017). This final algorithm, MC-LAVE-RL, alternates between planning via MC-LAVE and supervised learning of self-generated language actions. Experimental results demonstrate that MC-LAVE-RL achieves new high scores in various IF games provided in the Jericho framework (Hausknecht et al., 2020), showing the effectiveness of language-grounded MC-LAVE planning.

## 2 BACKGROUND

### 2.1 INTERACTIVE FICTION GAME

Interactive Fiction (IF) games are fully text-based environments where the observation and the action spaces are defined as natural language. The game-playing agent observes textual descriptions of the world, selects a language-based action, and receives the associated reward. IF games can be modeled as a special case of partially observable Markov decision processes (POMDPs) defined by tuple $\langle S, A, \Omega, T, O, R, \gamma \rangle$, where $S$ is the set of environment states $s$, $A$ is the set of language actions $a$, $\Omega$ is the set of text observations $o$, $T(s'|s,a) = \Pr(s_{t+1} = s'|s_t = s, a_t = a)$ is the transition function, $R(s,a)$ is the reward function for taking action $a$ in state $s$, $O(s) = o$ is the deterministic observation function in state $s$, and $\gamma \in (0, 1)$ is the discount factor. The history at time step $t$, $h_t = \{o_0, a_0, \ldots, o_{t-1}, a_{t-1}, o_t\}$, is a sequence of observations and actions. The goal is to find an optimal policy $\pi^*$ that maximizes the expected cumulative rewards, i.e. $\pi^* = \arg\max_\pi \mathbb{E}_\pi \left[ \sum_{t=0}^\infty \gamma^t R(s_t, a_t) \right]$.

We use the same definition of observation and action space as Hausknecht et al. (2020); Ammanabrolu & Hausknecht (2020); Côté et al. (2018), i.e. An observation is defined by $o_t = (o_{t_{\mathrm{desc}}}, o_{t_{\mathrm{game}}}, o_{t_{\mathrm{inv}}}, a_{t-1})$ where $o_{t_{\mathrm{desc}}}$ is the textual description of the current location of the agent, $o_{t_{\mathrm{game}}}$ is the simulator response to the previous action taken by the agent, $o_{t_{\mathrm{inv}}}$ is the information of agent's inventory, and $a_{t-1}$ is the previous action taken by the agent. An action is denoted by a sequence of words $a_t = (a_t^1, a_t^2, \ldots, a_t^{|a_t|})$. Finally, we denote $A_{\mathrm{valid}}(o_t) \subseteq A$ as the set of valid actions for the observation $o_t$, which is provided by the Jericho environment interface. Figure 1 shows an example of observation and action in ZORK1, one of the representative IF games.

### 2.2 CHALLENGES IN INTERACTIVE FICTION GAME

IF games pose important challenges for reinforcement learning agents, requiring natural language understanding, commonsense reasoning, and non-myopic language-grounded planning ability in combinatorial search space of language actions (Hausknecht et al., 2020). More concretely, consider the particular game state of ZORK1 described in Figure 1. In this example situation, a human player would be naturally capable of performing strategic planning via language understanding and commonsense reasoning: (1) the 'closed trap door' will have to be opened and be explored to proceed with the game, (2) however, acquiring the 'lantern' should precede entering the trap door since the cellar, which is expected to exist below the trap door, could be likely pitch-dark. Without such language-grounded reasoning and planning, the agent may need to try out every actions uniformly, most of them making it vulnerable to being eaten by a monster in the cellar who always appears when there is no light source. As a result, any agent that lacks the ability of long-term planning with language reasoning is prone to be stuck at a suboptimal policy, which enters the cellar to obtain an immediate reward and does nothing further to avoid encountering the monster that kills the agent immediately.

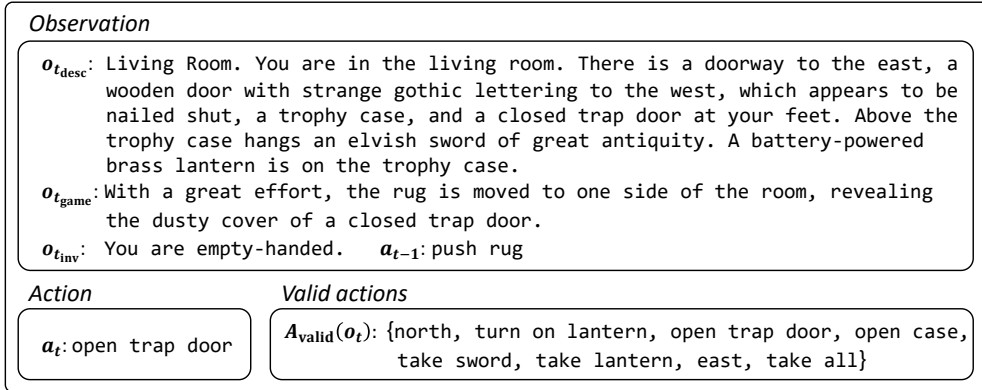

Figure 1: Example of observation and actions from ZORK1

To overcome the bottleneck, the agent should be able to infer that 'take lantern' is worthy enough to be chosen in preference to other actions even though it gives no immediate reward. In this work, we will precisely address such failures of myopic planning that arise from the lack of language-grounded exploration strategy. To this end, we first propose MCTS-based long-term planning for nonmyopic-planning. Then we further improve the search efficiency of MCTS-based planning by incorporating our novel language-grounded exploration strategy based on Language Action Value Estimates, which will be detailed in the subsequent section.

## 2.3 MONTE-CARLO TREE SEARCH

Monte-Carlo Tree Search (MCTS) (Kocsis & Szepesvári, 2006; Coulom, 2006; Browne et al., 2012) is a generic online planning algorithm that combines random sampling and tree search, which has become the de-facto standard method for large sequential decision-making problems. Starting from an empty tree, MCTS repeatedly performs the phases of selection, expansion, rollout, and backpropagation to evaluate the nodes of the search tree with increased accuracy. UCT (Kocsis & Szepesvári, 2006) is the standard MCTS method, which adopts UCB (Auer et al., 2002) as an action selection rule at each internal node of the search tree:

$$\arg\max_a \left[ Q(h, a) + c \sqrt{\frac{\log N(h)}{N(h, a)}} \right] \tag{1}$$

where $h$ is the history of past observations and actions, $Q(h, a)$ is the average value of sampled returns with taking action $a$ in $h$, $N(h)$ is the number of simulations performed in $h$, $N(h, a)$ is the number of times action $a$ is selected in $h$, and $c$ is a constant that balances exploration and exploitation. However, UCT suffers from severe search inefficiency in the problems with a large action space such as IF games, since it requires to take every action at least once and relies only on *uncertainty-driven* (or visit-count-based) exploration.

PUCT (Silver et al., 2017; 2018) partially addresses the challenges of IF games by adopting PUCB (Rosin, 2011), which involves a prior action distribution $\pi(\cdot|h)$ and eliminates the need for choosing every action at least once. The action selection rule is given by:

$$\arg\max_a \left[ Q(h, a) + c_{\text{PUCT}} \pi(a|h) \frac{\sqrt{N(h)}}{N(h, a) + 1} \right] \tag{2}$$

The prior distribution $\pi(\cdot|h)$ in Eq. (2) can be trained by behavioral cloning of $(h_{\text{root}}, a^*_{\text{root}})$ samples obtained by the result of tree search in previous time steps (Anthony et al., 2017; Silver et al., 2018). However, this procedure not only discards other search information such as abundant $(h_t, a_t, r_t, h_{t+1})$ samples obtained during the search but also hardly encourages information sharing across the semantically similar actions. Therefore, PUCT is still limited in search efficiency for the tasks with language action space, raising the need for a *language-driven* exploration strategy.

## 3 MONTE-CARLO PLANNING WITH LANGUAGE ACTION VALUE ESTIMATES

Language-grounded planning is essential to address the challenges of IF games. To this end, we need a mechanism to incorporate language understanding and commonsense reasoning into the ex-

ploration strategy of search. In this section, we introduce Monte-Carlo planning with Language Action Value Estimates (MC-LAVE), a novel planning algorithm that combines MCTS with language-driven exploration. MC-LAVE does not rely on any handcrafted heuristic evaluation but starts tabula rasa, based only upon pre-trained word embeddings that can well define the semantic similarities between two language actions. In addition, it only requires a black-box simulator $(r, s', o') \sim \mathcal{G}(s, a)$, which yields $(o, a, r, o')$ samples at each simulator query.

## 3.1 Credit Assignment with Search Experience

Language-driven exploration in MC-LAVE starts by identifying which word sequences of language action are likely to yield large long-term rewards. This credit assignment of language action is done by Q-learning using all the experiences $\mathcal{D} = \{(o, a, r, o')\}$ obtained during the past searches. Following the previous work (Hausknecht et al., 2020; Ammanabrolu & Hausknecht, 2020; Ammanabrolu et al., 2020; Madotto et al., 2020), we assume that the current observation captures all the information regarding the latent state, as it is just a text description of the game state. Therefore, we represent the Q-function as a function of the current observation and action, rather than a function of full history and action. Our parameterization of $Q_\phi(o, a)$ basically follows that of Deep Reinforcement Learning Relevance Network (DRRN) (He et al., 2016; Hausknecht et al., 2020), and more details can be found in Appendix A. Finally, the $Q_\phi$ is trained by minimizing the temporal difference error using $\mathcal{D}$:

$$\arg\min_{\phi} \mathbb{E}_{(o,a,r,o')\sim\mathcal{D}} \left[ \left( r + \gamma \max_{a' \in A_{\text{valid}}(o')} Q_{\bar{\phi}}(o', a') - Q_\phi(o, a) \right)^2 \right] \tag{3}$$

where $\bar{\phi}$ is an exponential moving average of $\phi$, i.e. soft target update, and $A_{\text{valid}}(o')$ is a set of valid actions in the observation $o'$ provided by Jericho environment. While PUCT of Eq. (2) also uses previous search information to construct a prior policy $\pi$, it is typically trained only on planning trajectories $(h_{\text{root}}, a_{\text{root}}^*)$ that generally cover very limited regions. Therefore, it can hardly provide informative search guides for areas outside the planning trajectories. In contrast, $Q_\phi$ of Eq. (3) is trained by using every search experience encountered during search processes, thus it can potentially provide more information over a much broader set of states and actions. However, the credit assignment provided by $Q_\phi$ still struggles to provide useful information for novel language observation-action regions that have never been experienced, making its direct adoption as an exploration strategy inappropriate: the search process necessarily entails visits to novel areas where $Q_\phi$ does not work properly, and thus $Q_\phi$ may instead promote exploitation.

## 3.2 Language-driven Exploration via Language Action Value Estimates

In order to generalize past experiences to novel situations, we adopt context-agnostic information sharing between language actions based on their semantic similarities. For example, if an action of 'take something' has ever been useful in the past, we may expect that the action of 'take something else' can also be worthy in other situations. Motivated by such instances, we introduce Language Action Value Estimates (LAVE), which assigns *non-uniform* search priorities to each language action by examining whether its semantic neighborhood actions have been beneficial in the past.

As the first step for LAVE, we define a *semantic neighborhood* of action $a$, a set of the past experiences whose actions share similar semantics with action $a$ by:

$$\mathcal{N}(a) = \{(\bar{o}, \bar{a}) \mid d(a, \bar{a}) < \delta \text{ where } (\bar{o}, \bar{a}, \bar{r}, \bar{o}') \in \mathcal{D}\} \tag{4}$$

where $\delta$ is a hyperparameter that controls a proximity threshold, and $d(a, \bar{a})$ is a distance measure between two actions $a$ and $\bar{a}$ in language embedding space represented by the pre-trained word embeddings $e(\cdot)$:

$$d(a, \bar{a}) = 1 - \frac{\psi(a) \cdot \psi(\bar{a})}{\|\psi(a)\|\|\psi(\bar{a})\|} \text{ where } \psi(a) = \frac{1}{|a|} \sum_{i=1}^{|a|} e(a^i)$$

For simplicity, we use the average of the word embeddings $\psi(a)$ as the language action (or sentence) embedding, where the semantic similarity can be well-measured by the cosine similarity in the embedding space.

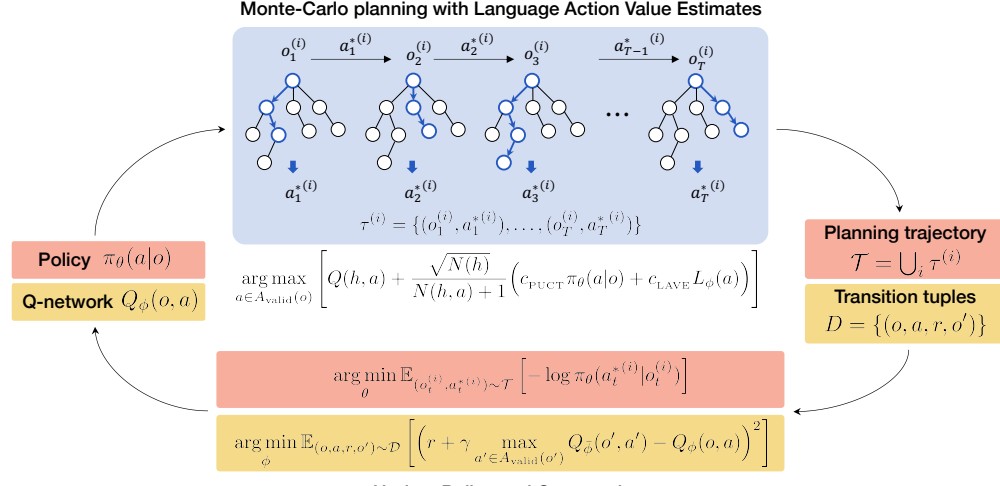

Figure 2: Overall architecture of policy improvement via MC-LAVE.

Then, the LAVE exploration bonus for the action $a$ is estimated through a soft maximum over the Q-values of $a$'s semantic neighborhood actions in order to encourage optimistic exploration.

$$L_\phi(a) = \log \frac{1}{|\mathcal{N}(a)|} \sum_{(\bar{o},\bar{a}) \in \mathcal{N}(a)} \exp[Q_\phi(\bar{o},\bar{a})] \tag{5}$$

Note that $L_\phi(a)$ in Eq. (5) relies only on the Q-values of $(\bar{o}, \bar{a})$ samples experienced in the past, thus can be reliably estimated. Also, it is context-agnostic in which the action $a$ is taken, thus circumvents the challenge of measuring the similarity between the very long textual observations, while promoting active information sharing between affordable language actions.

Finally, MC-LAVE encourages the *language-driven* exploration in the selection phase of MCTS via LAVE $L_\phi(a)$, which selects actions at each intermediate nodes by the following rule:

$$\underset{a \in A_{\text{valid}}(o)}{\arg\max} \left[ Q(h,a) + \underbrace{\frac{\sqrt{N(h)}}{N(h,a)+1}}_{\substack{\text{uncertainty-driven} \\ \text{exploration}}} \left( \underbrace{c_{\text{PUCT}} \pi_\theta(a|o)}_{\substack{\text{prior knowledge} \\ \text{encoding}}} + \underbrace{c_{\text{LAVE}} L_\phi(a)}_{\substack{\text{language-driven} \\ \text{exploration}}} \right) \right] \tag{6}$$

where $\pi_\theta(a|o)$ is a prior policy trained by behavior cloning of past planning results, $L_\phi(a)$ is a language-driven exploration bonus, $c_{\text{LAVE}}$ is an exploration constant. MC-LAVE uses the policy $\pi_\theta$ for incorporating prior knowledge of the past planning trajectories, as well as the language-driven bonus $L_\phi(a)$ for facilitating information sharing between semantic neighborhood actions. MC-LAVE initially encourages to explore the actions with high language-driven exploration bonuses and low visit count, but it does not hurt the asymptotic behavior of MCTS (Theorem 2 in Appendix C). The pseudo-code for the MC-LAVE is provided in Appendix D.

## 4 REINFORCEMENT LEARNING VIA MC-LAVE

As shown in (Silver et al., 2017), MCTS can be used as a strong policy improvement operator for reinforcement learning. Similarly, we use MC-LAVE as a policy improvement operator for learning the IF game. As a result, our reinforcement learning algorithm, MC-LAVE-RL, alternates between MC-LAVE planning and supervised learning of self-generated actions:

1. **Planning** Use the MC-LAVE with current policy $\pi_\theta$ and Q-function $Q_\phi$ to collect the $n$ planning trajectories $\mathcal{T} = \bigcup_i \tau^{(i)}$ and experience replay $\mathcal{D}$:

$$\tau^{(i)} = \{(o_1^{(i)}, a_1^{*(i)}), \ldots, (o_T^{(i)}, a_T^{*(i)})\}, \mathcal{D} = \{(o, a, r, o')\}$$

where $a_t^{*(i)}$ is selected action at $i$-th trajectory $t$-th timestep by MCTS and $T$ denotes an episode length. (For the first iteration, MC-LAVE uses uniform policy and randomly initialized Q-function.)

| | Algorithms | DRRN[†] | TDQN[†] | KG-A2C[‡] | MC!Q*BERT[§] | PUCT-RL | MC-LAVE-RL |
|---|---|---|---|---|---|---|---|
| **Require-ments** | *Valid Action* | Hard | Soft | Soft | Soft | Hard | Hard |
| | *Resettable* | - | - | - | ✓ | ✓ | ✓ |
| | *External Info* | - | - | - | ALBERT & Jericho-QA | - | Word Embedding |
| **Difficult Games** | ZORK1 | 32.6 | 9.9 | 34 | 41.6 | 38.2 | **45.2** |
| | DEEPHOME | 1 | 1 | 1 | 8 | 28.6 | **35** |
| | LUDICORP | 13.8 | 6 | 17.8 | **22.8** | 18 | **22.8** |
| **Possible Games** | PENTARI | 27.2 | 17.4 | 50.7 | 58 | 64 | **68** |
| | DETECTIVE | 197.8 | 169 | 207.9 | **330** | 322 | **330** |
| | LIBRARY | 17 | 6.3 | 14.3 | **19** | **19** | **19** |
| | BALANCES | **10** | 4.8 | **10** | **10** | **10** | **10** |
| | TEMPLE | 7.4 | 7.9 | **8** | **8** | **8** | **8** |
| | ZTUU | **21.6** | 4.9 | 9.2 | 11.8 | 5 | 7 |

Table 1: Performance evaluation of various approaches for IF games in Jericho (Hausknecht et al., 2020). Handicaps leveraged in each algorithm are described in the table. *Valid Action* indicates whether an algorithm uses a valid action handicap provided by Jericho as a hard constraint for the effective action space or as a soft constraint via entropy loss over the valid actions (Ammanabrolu & Hausknecht, 2020). *Resettable* indicates whether an algorithm requires a resettable simulator, i.e. the previously visited state can be restored. ZORK1, DEEPHOME, LUDICORP are categorized as *difficult games* and the other games are categorized as *possible games* by Hausknecht et al. (2020). The results of other algorithms are from Hausknecht et al. (2020)[†], Ammanabrolu & Hausknecht (2020)[‡], and Ammanabrolu et al. (2020)[§] respectively. All our results (PUCT-RL, MC-LAVE-RL) indicate averages over 5 independent runs, and their standard errors are provided in Appendix E due to space limit.

2. **Policy training** Update the policy network parameter $\theta$ by minimizing the negative log-likelihood with using collected planning trajectories $\mathcal{T}$, i.e. supervised learning with cross-entropy loss:

$$\arg\min_{\theta} \mathbb{E}_{(o_t^{(i)}, a_t^{*(i)}) \sim \mathcal{T}} \left[ -\log \pi_\theta(a_t^{*(i)} | o_t^{(i)}) \right]$$

3. **Q-learning** Update the Q-function parameter $\phi$ by minimizing the temporal difference error from experience replay $\mathcal{D}$:

$$\arg\min_{\phi} \mathbb{E}_{(o,a,r,o') \sim \mathcal{D}} \left[ \left( r + \gamma \max_{a' \in A_{\text{valid}}(o')} Q_{\bar{\phi}}(o', a') - Q_\phi(o, a) \right)^2 \right]$$

We continue to update the policy through the above policy iteration with the MC-LAVE operator until the policy performance converges.

## 5 EXPERIMENTS

In this section, we show experimental results of our approach on IF games included in the Jericho environment (Hausknecht et al., 2020). The action spaces of IF games in the Jericho environment are defined by the sequences of the game-dependent input vocabulary whose size is roughly from 500 to 2000. While the size of the vocabulary is limited compared to large-scale open domain dialogues, it still serves as a useful testbed to demonstrate the effectiveness of semantic-based information sharing of MC-LAVE[1]. First, we evaluate the performances of MC-LAVE-RL comparing with baseline methods. Second, we compare intermediate scores of MC-LAVE-RL on ZORK1 with a baseline (PUCT-RL) that uses PUCT as a policy improvement operator, to show that MC-LAVE is more effective for IF games than PUCT. Third, we investigate the effectiveness of semantic neighborhood by comparing the planning performance of MC-LAVE under the various sizes of the semantic neighborhood. Finally, we give a qualitative analysis of our approach using a representative example, which shows how MC-LAVE focuses on semantically promising actions during the search. Detailed configurations of our experiments are described in Appendix B.

---

[1]For example, the ZORK1 environment produces identical observations for taking the actions 'take lantern' and 'get lantern', where MC-LAVE can readily take advantage by exploiting the information sharing across those semantically similar actions (Appendix H).

| Methods | PUCT-RL | | MC-LAVE-RL | |
|---|---|---|---|---|
| | Planning | Learning | Planning | Learning |
| Iteration 1 | $31.9 \pm 1.4$ | $36.6 \pm 1.0$ | $30.4 \pm 2.0$ | $36.6 \pm 1.0$ |
| Iteration 2 | $35.8 \pm 0.0$ | $37.4 \pm 1.0$ | $36.1 \pm 0.1$ | $38.2 \pm 0.8$ |
| Iteration 3 | $35.3 \pm 0.2$ | $39.0 \pm 0.0$ | $41.2 \pm 0.5$ | $43.0 \pm 1.0$ |
| Iteration 4 | $35.2 \pm 0.4$ | $38.2 \pm 0.8$ | $43.8 \pm 0.1$ | $45.2 \pm 1.2$ |

Table 2: Experimental results of the policy improvement with PUCT and MC-LAVE on ZORK1. For each iteration, we perform the 25 planning with PUCT and MC-LAVE to collect planning trajectories and experience replay. All results indicate averages and standard errors over 5 trials.

## 5.1 EVALUATION ON INTERACTIVE FICTION GAMES

First, we compare the performance of MC-LAVE-RL with the following algorithms: (1) DRRN (Hausknecht et al., 2020), a variant of the DQN algorithm (Mnih et al., 2013) for natural language action space, (2) TDQN (Hausknecht et al., 2020), an extension of LSTM-DQN algorithm (Narasimhan et al., 2015) incorporating with template-based action generation, (3) KG-A2C (Ammanabrolu & Hausknecht, 2020), an actor-critic method with knowledge graph state representation, (4) MC!Q*BERT (Ammanabrolu et al., 2020), an extension of KG-A2C with BERT-based knowledge graph construction and knowledge-graph-based intrinsic reward. In addition, we also compare MC-LAVE-RL with our baseline called PUCT-RL, which uses PUCT as a policy improvement operator.

Table 1 summarizes handicaps leveraged in each algorithm and the performance of MC-LAVE-RL and baseline algorithms across 9 IF games included in the Jericho environment. The results show that MC-LAVE-RL outperforms or matches the state-of-the-art results on 8 out of 9 games. Although MC-LAVE-RL requires more handicap or assumption, it performs the same or better than strong baseline MC!Q*BERT which requires similar assumptions and more requirements. In addition, MC-LAVE-RL achieves higher game scores on overall games compared to PUCT-RL, which is a baseline algorithm that only excludes language-driven exploration strategy from MC-LAVE-RL. Furthermore, MC-LAVE-RL performs significantly better than other methods on *difficult games* such as ZORK1, DEEPHOME, and LUDICORP, which are categorized by Hausknecht et al. (2020) as a relatively challenging game due to the large action space and sparse rewards.

In the case of ZORK1, as described in Section 2.2, other algorithms (except MC!Q*BERT, which is concurrent work by Ammanabrolu et al. (2020)) fail to overcome the bottleneck and are stuck into the suboptimal policy, which leaves the agent being eaten by a monster in the dark without a light source. On the other hand, MC-LAVE-RL successfully finds the optimal action (i.e. 'take lantern') on the bottleneck state by using the exploration strategy suitable for natural language spaces and obtains additional rewards by discovering novel states that cannot be reached with other methods. For instance, trajectories of MC-LAVE-RL agent frequently show that the agent overcomes bottlenecks and behaves crucial actions to complete the game, such as 'killing the monster with sword' and 'putting the painting in the trophy case'. Further illustrative examples of trajectories are provided in Appendix F.

## 5.2 REINFORCEMENT LEARNING VIA MC-LAVE

In order to understand the effectiveness of MC-LAVE as a policy improvement operator, we compare the performances of PUCT-RL and MC-LAVE-RL in ZORK1. Table 2 reports the intermediate results of planning and supervised learning in each iteration of the policy iteration. In each iteration, the policy and the Q-function are trained using planning trajectories and experience replay collected from 25 independent planning agents. As can be seen in Table 2, the performance of MC-LAVE-RL is improved more consistently than PUCT-RL, both in planning and learning. At the beginning of the policy iteration, PUCT-RL improves the performance, but it fails to overcome bottleneck and converges to a suboptimal policy: PUCT utilizes the prior policy learned by imitating the planning results of the previous iteration to estimate the exploration bonus, but this uncertainty-based method is not much effective to encourage the agent to explore the action space that is not sufficiently covered. On the other hand, MC-LAVE-RL not only uses the prior policy, but also uses Q-Network for credit assignment to language actions. This allows a more focused exploration on semantically promising actions and consequently overcomes the bottleneck to further improve the performance.

| PUCT | east | open case | open trap door | take all | take lantern | take sword | turn on lantern |
|---|---|---|---|---|---|---|---|
| $Q(h,a)$ | 2.65 | 6.80 | **13.81** | 2.03 | 11.34 | 0.22 | 9.00 |
| $\pi_\theta(a\|o)$ | 0.08 | 0.08 | 0.50 | 0.08 | 0.08 | 0.08 | 0.08 |
| $N(h,a)$ | 38 | 39 | 105 | 37 | 52 | 37 | 42 |

| MC-LAVE | east | open case | open trap door | take all | take lantern | take sword | turn on lantern |
|---|---|---|---|---|---|---|---|
| $Q(h,a)$ | 7.86 | 10.59 | 14.10 | 2.30 | **14.23** | -1.33 | 13.11 |
| $\pi_\theta(a\|o)$ | 0.08 | 0.08 | 0.50 | 0.08 | 0.08 | 0.08 | 0.08 |
| $N(h,a)$ | 23 | 58 | 85 | 47 | 67 | 7 | 63 |
| $L_\phi(a)$ | 13.21 | 35.55 | 25.35 | 38.21 | 49.81 | 2.25 | 34.72 |

Table 3: Illustrative examples of search results of PUCT and MC-LAVE on bottleneck state in ZORK1 (see Figure 1). $Q(h,a)$ denotes the average of Monte-Carlo returns, $\pi_\theta(a|o)$ represents policy prior, $N(h,a)$ represents visit count of each action, $L_\phi(a)$ represents language-driven exploration bonus of MC-LAVE. $c_{\text{PUCT}} = 200$ and $c_{\text{LAVE}} = 0.1$ are used for the exploration constants.

### 5.3 THE EFFECT OF THE SIZE OF SEMANTIC NEIGHBORHOOD

The size of semantic the neighborhood is particularly crucial for the performance of MC-LAVE planning. To see this, we conducted a simple ablation experiment in ZORK1 on the effect of varying proximity threshold $\delta$, which controls the size of the semantic neighborhood of action $\mathcal{N}(a)$ in Eq. (4). Moreover, to demonstrate that the LAVE exploration bonus works as a meaningful exploration bonus beyond simple random noise injection, we also present the result of a simple baseline PUCT+RANDOM, which uses MC-LAVE action selection rule of Eq. (6) with the neighborhood given by 10 randomly sampled actions. This baseline can be understood as performing the search with noise injection via the randomly chosen neighborhood, while not considering information sharing between semantically similar actions. As shown in Figure 3, the performance of PUCT+RANDOM is almost identical to PUCT, which

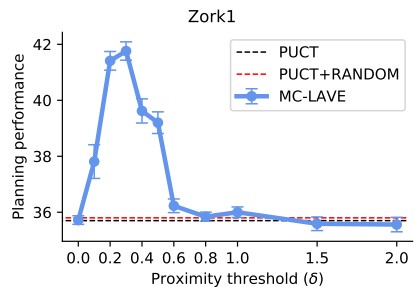

Figure 3: Performance of MC-LAVE planning on ZORK1 for varying $\delta$. The results are averaged over 100 trials, and error bars indicate the standard errors.

supports that random noise injection alone does not lead to improving search efficiency. Moreover, the results of MC-LAVE shows that too small or too large $\delta$ are not effective, which can be naturally explained as follows: (1) when $\delta = 0$, no sample in the replay buffer is considered as neighborhood, leading LAVE exploration bonus to always zero, (2) when $\delta = 2$ (the largest possible value), every sample in the replay buffer is considered as neighborhood, yielding LAVE exploration bonus to a constant value for all $a$. Therefore, with either two extremes of $\delta$, the MC-LAVE action selection rule of Eq. (6) is reduced to the PUCT action selection rule of Eq. (2). Finally, the moderately small $\delta = 0.3$ works the best, which highlights the importance of information sharing between *meaningfully defined* semantic neighborhoods to solve IF games efficiently.

### 5.4 QUALITATIVE ANALYSIS

In this section, we provide a qualitative analysis on the effectiveness of language-driven exploration in MC-LAVE. We show an illustrative example in Table 3 that demonstrates how MC-LAVE focuses more on the semantically promising actions, comparing the search results of PUCT and MC-LAVE in the bottleneck state of ZORK1 described in Figure 1. In this example, the 'open trap door' is a suboptimal action that obtains an immediate reward, while 'take lantern' gives no immediate reward, but it is an optimal action when considering long-term outcome. As can be seen in Table 3, PUCT cannot explore effectively because the search is mostly based on uncertainty, and converges to suboptimal action (i.e. 'open trap door') that obtains an immediate reward. On the other hand, MC-LAVE assigns search priorities to actions based on the optimistically aggregated Q-estimates of the past actions. As the result of focusing on semantically promising actions, MC-LAVE selects the optimal action (i.e. 'take lantern') even though it does not give any immediate reward. PUCT also performs non-uniform search based on prior policy, but this only encourages to explore the actions of previous behavior and does not focus on semantically promising actions.

## 6    RELATED WORK

**Variants of Monte-Carlo tree search.** Leveraging prior knowledge for MCTS has been one of the most important directions for improving MCTS (Gelly & Silver, 2007; Silver et al., 2016; 2017; 2018; Gelly & Silver, 2011). PUCT (Rosin, 2011; Silver et al., 2017) is one of the most popular MCTS algorithms, which incorporates prior knowledge as a policy. The prior policy in PUCT is typically trained by imitation learning of past behavior; other search information, such as transition tuples and inferred Q-values, is discarded. To address this limitation, Hamrick et al. (2019) present Search with Amortized Value Estimates (SAVE), which incorporates Q-based prior knowledge into MCTS. In SAVE, Q-function is fit by inferred Q-values and transition tuples from search, then is used as a prior into MCTS. Although SAVE improves sample efficiency in MCTS, it lacks a language-grounded exploration strategy. Rapid Action Value Estimation (RAVE) (Gelly & Silver, 2011) is another notable method that addresses information sharing, which allows sharing action values in the search tree. However, RAVE is designed especially for the game of Go and cannot be straightforwardly applied to IF games. Furthermore, to leverage the prior knowledge given as textual information, Branavan et al. (2011) present a method that extracts the most relevant information to the current state and incorporates linguistic information into the Monte-Carlo search framework. However, they assume natural language documents that contain domain knowledge of the task.

**Reinforcement Learning via Monte-Carlo tree search.** There are a number of recent works on using MCTS for reinforcement learning. Silver et al. (2017) achieved successful results with MCTS as a policy improvement operator in the game of Go. Also, in the negotiation dialogue domain, Jang et al. (2020) used Bayes-adaptive Monte-Carlo planning as a policy improvement operator to prevent the issue of diverging from human language. Inspired by these successes, we use the MC-LAVE algorithm as a policy improvement operator which effectively works on IF games.

**Deep Reinforcement Learning for Interactive Fiction Games.** IF games have attracted a lot of interest from the RL research since the introduction of Jericho (Hausknecht et al., 2020), an OpenAI Gym-like (Brockman et al., 2016) learning environment for IF games. Jericho includes baseline RL agents such as DRRN and TDQN. DRRN is a variant of deep Q-Network architecture for handling natural language actions in text-based decision making tasks. TDQN is an extension of LSTM-DQN (Narasimhan et al., 2015) utilizing predefined templates in order to reduce effective search spaces of combinatorial language action space. More recently, knowledge-graph-based approaches have been introduced to address the issues of partial observability and commonsense reasoning (Ammanabrolu & Riedl, 2019; Ammanabrolu & Hausknecht, 2020; Ammanabrolu et al., 2020). KG-A2C (Ammanabrolu & Hausknecht, 2020) is an actor-critic algorithm that leverages knowledge graph state representation for commonsense reasoning. Ammanabrolu et al. (2020) introduce Q*BERT, an extension of KG-A2C, which updates the knowledge graph by using the pre-trained language model ALBERT (Lan et al., 2020), a variant of BERT (Devlin et al., 2019). To encourage better exploration and overcome the bottleneck state in IF games, they introduce MC!Q*BERT that additionally uses intrinsic motivation based on changes of the knowledge graphs.

## 7    CONCLUSION

We presented Monte-Carlo planning with Language Action Value Estimates (MC-LAVE), an MCTS-based algorithm for language action space, which combines MCTS with language-driven exploration strategy. MC-LAVE assigns search priorities to each language action based on Q-values of previously executed actions that share similar meanings, then invests more search effort into semantically promising actions. We then incorporated MC-LAVE into reinforcement learning by using MC-LAVE as a policy improvement operator. Our approach achieved remarkable results in difficult games that feature large action space and sparse reward and outperforms or matches the state-of-the-art on the various IF games.

ACKNOWLEDGMENTS

This work was supported by the National Research Foundation (NRF) of Korea (NRF-2019M3F2A1072238 and NRF-2019R1A2C1087634), the Ministry of Science and Information communication Technology (MSIT) of Korea (IITP No. 2019-0-00075, IITP No. 2020-0-00940 and IITP No. 2017-0-01779 XAI), and the ETRI (Contract No. 20ZS1100).

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

## A    IMPLEMENTATION DETAILS

As discussed in section 4, MC-LAVE-RL alternates (1) planning, (2) supervised learning and Q-learning. In the planning phase, multiple planning workers are run in parallel on a distributed computing system to collect various planning trajectories and experience replay. After planning, we merge collected trajectories and experience replay and train the policy network and Q-Network using merged trajectories and experience replay respectively. For Q-Network, we exploit the same architecture used in DRRN (He et al., 2016) illustrated as Figure 5, which takes an observation $o_t$, an action $a_t$ as an input and outputs $Q_\phi(o_t, a_t)$. For the policy network, we use a network architecture similar with DRRN as described in Figure 4, which directly outputs $f_\theta(a_t|o_t)$, an unnormalized probability of an action $a_t$ given an observation $o_t$, i.e. $\pi_\theta(a_t|o_t) \propto f_\theta(a_t|o_t)$. Our code is publicly available[2].

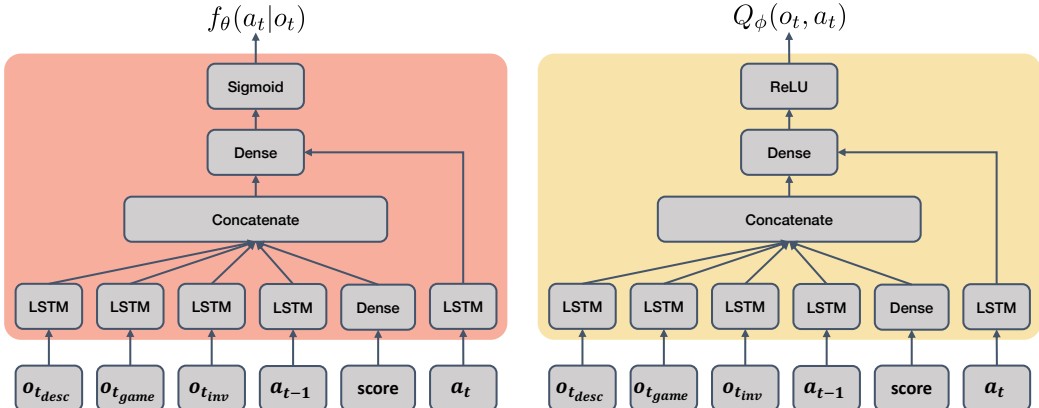

Figure 4: Policy network          Figure 5: Q-Network

## B    EXPERIMENTS DETAILS

| Hyperparameters | ZORK1 | DEEPHOME | LUDICORP | PENTARI | DETECTIVE | LIBRARY | BALANCES | TEMPLE | ZTUU |
|---|---|---|---|---|---|---|---|---|---|
| $\lambda$ (discount factor) | 0.95 | 0.95 | 0.95 | 0.95 | 0.95 | 0.95 | 0.95 | 0.95 | 0.95 |
| # of max iterations | 4 | 4 | 4 | 4 | 4 | 4 | 4 | 4 | 4 |
| $\delta$ (neighborhood threshold) | 0.3 | 0.3 | 0.3 | 0.3 | 0.3 | 0.3 | 0.3 | 0.3 | 0.3 |
| Max search depth | 10 | 10 | 20 | 10 | 10 | 15 | 15 | 10 | 15 |
| Max episode length | 35 | 35 | 50 | 35 | 50 | 35 | 35 | 35 | 35 |
| # of simulations per action | 50 | 50 | 50 | 50 | 50 | 50 | 50 | 50 | 50 |
| # of planning workers | 25 | 25 | 25 | 25 | 25 | 25 | 25 | 25 | 25 |
| $c_{\text{LAVE}}$ | 1.0 | 1.0 | 2.0 | 1.0 | 0.1 | 1.0 | 1.0 | 1.0 | 1.0 |
| $c_{\text{PUCT}}$ | 50 | 20 | 50 | 50 | 200 | 20 | 50 | 50 | 50 |

Table 4: Configurations of MC-LAVE-RL used in our experimental results. Hyperparameters in the upside of the table were globally adapted in the planning-learning framework and the other hyperparameters are used only in the MCTS planning phase.

---

[2]https://github.com/jys5609/MC-LAVE-RL

## C    ASYMPTOTIC OPTIMALITY OF MC-LAVE

In this section, we show that the asymptotic optimality of PUCB (Rosin, 2011) is still held after the introduction of the language-driven exploration bonus in Eq. (5). We start by restating the theoretical result for PUCB.

**Lemma 1.** *(Theorem 1 in (Rosin, 2011)) For any stochastic policy $\pi(a) > 0$ with $\sum_a \pi(a) = 1$, $c_1 > 0$, and $c_2 > 0$, **PUCB action selection rule** at trial $t$ is defined as:*

$$\arg\max_a \left[ r(t,a) + c_1 \sqrt{\frac{\log t}{N(a)}} - c_2 \frac{1}{\pi(a)} \sqrt{\frac{\log t}{t}} \right]$$

*where $N(a)$ is the number of simulations action $a$ is taken in the past, $r(t,a)$ is an empirical average reward of action $a$ at the start of trial $t$., Then, expected regret of PUCB action selection rule is bounded by $O(\sqrt{n \log n})$ $\forall n > 1$.*

Note that the prior policy $\pi$ in the PUCB rule encourages the actions that have a high value of $\pi(a)$ to be selected more often. In this work, we use a variant of PUCB as in (Silver et al., 2017).

**Theorem 2.** *MC-LAVE action selection rule at trial $t$ is defined as:*

$$\arg\max_a \left[ r(t,a) + c_1 \sqrt{\frac{\log t}{N(a)}} - c_2 \frac{1}{\pi(a)} \sqrt{\frac{\log t}{t}} - c_3 \frac{1}{L(a)} \sqrt{\frac{\log t}{t}} \right]$$

*where $N(a)$ is the number of simulations action $a$ is taken in the past, $r(t,a)$ is an empirical average reward of action $a$ at the start of trial $t$, $\pi(a) > 0$ is a stochastic policy with $\sum_a \pi(a) = 1$, $c_1 > 0$, $c_2 > 0$, $c_3 > 0$ and $L(a) > 0$ is a bonus of action $a$. Then MC-LAVE action selection rule also achieves expected regret bounded by $O(\sqrt{n \log n})$ $\forall n > 1$.*

*Proof.* The theorem can be proved by showing that the MC-LAVE action selection rule can be represented as the PUCB action selection rule with another policy. MC-LAVE action selection rule is defined as

$$\arg\max_a \left[ r(t,a) + c_1 \sqrt{\frac{\log t}{N(a)}} - c_2 \frac{1}{\pi(a)} \sqrt{\frac{\log t}{t}} - c_3 \frac{1}{L(a)} \sqrt{\frac{\log t}{t}} \right]$$

$$= \arg\max_a \left[ r(t,a) + c_1 \sqrt{\frac{\log t}{N(a)}} - c_2 \left( \frac{1}{\pi(a)} + \frac{c_3}{c_2} \frac{1}{L(a)} \right) \sqrt{\frac{\log t}{t}} \right]$$

Define $p(a)$, $\pi'(a)$, $c_2'$ as follows:

$$p(a) := \frac{1}{\left( \frac{1}{\pi(a)} + \frac{c_3}{c_2} \frac{1}{L(a)} \right)}, \quad \pi'(a) := \frac{p(a)}{\sum_a p(a)}, \quad c_2' := c_2 \frac{1}{\sum_a p(a)}$$

Then MC-LAVE action selection rule can be represented as follows:

$$\arg\max_a \left[ r(t,a) + c_1 \sqrt{\frac{\log t}{N(a)}} - c_2 \frac{1}{p(a)} \sqrt{\frac{\log t}{t}} \right]$$

$$= \arg\max_a \left[ r(t,a) + c_1 \sqrt{\frac{\log t}{N(a)}} - c_2 \frac{1}{\sum_a p(a)} \cdot \frac{1}{\frac{p(a)}{\sum_a p(a)}} \sqrt{\frac{\log t}{t}} \right]$$

$$= \arg\max_a \left[ r(t,a) + c_1 \sqrt{\frac{\log t}{N(a)}} - c_2' \frac{1}{\pi'(a)} \sqrt{\frac{\log t}{t}} \right]$$

which induces PUCB action selection rule with another policy $\pi'$ and coefficient $c_2'$. Hence, by Lemma 1, MC-LAVE achieves expected regret bounded by the same order with PUCB, which is $O(\sqrt{n \log(n)})$ $\forall n > 1$.                                   □

Thus, in the same manner, we obtain the analysis of UCT from that of UCB, i.e. induction with a mild assumption on the drift-condition of the inherent non-stationary bandit problem, we can obtain the analysis of MCTS using MC-LAVE using the above result.

# D    PSEUDOCODE OF MC-LAVE

---

**Algorithm 1** Monte-Carlo Planning with Language Action Value Estimates (MC-LAVE)

---

**procedure** SEARCH($s_0$)
    $o_0 \leftarrow O(s_0)$
    $h_0 \leftarrow o_0$
    **repeat**
        SIMULATE $(s_0, h_0, 0)$
    **until** TIMEOUT ()
    **return** $\arg\max\limits_{a \in A_{\mathrm{valid}}(o_0)} Q(h_0, a)$
**end procedure**

**procedure** SIMULATE($s, h, t$)
    **if** $t = $ (planning horizon $H$) **then**
        **return** 0
    **end if**
    $[a, \mathrm{rollout}] \leftarrow$ SELECTACTION($h$)
    $[r, s', o'] \leftarrow \mathcal{G}(s, a)$
    $h' \leftarrow hao'$
    **if** rollout **then**
        $R' \leftarrow$ ROLLOUT($s', t + 1$)
    **else**
        $R' \leftarrow$ SIMULATE($s', h', t + 1$)
    **end if**
    $R \leftarrow r + \gamma \cdot R'$
    $N(h) \leftarrow N(h) + 1$
    $N(h, a) \leftarrow N(h, a) + 1$
    $Q(h, a) \leftarrow Q(h, a) + \frac{R - Q(h,a)}{N(h,a)}$
    **return** $R$
**end procedure**

**procedure** SELECTACTION($h$)

$$a \leftarrow \arg\max_{a \in A_{\mathrm{valid}}(o)} \left[ Q(h, a) + c_{\mathrm{PUCT}} \pi_\theta(a|o) \frac{\sqrt{N(h)}}{N(h,a)+1} \right.$$
$$\left. + c_{\mathrm{LAVE}} L_\phi(a) \frac{\sqrt{N(h)}}{N(h,a)+1} \right]$$

    **if** $N(h, a) = 0$ **then**
        rollout $\leftarrow$ true
    **else**
        rollout $\leftarrow$ false
    **end if**
    **return** $[a, \mathrm{rollout}]$
**end procedure**

**procedure** ROLLOUT($s, t$)
    **if** $t = $ (planning horizon $H$) **then**
        **return** 0
    **end if**
    $o \leftarrow O(s)$
    $a \sim \pi_\theta(\cdot|o)$
    $[r, s', o'] \leftarrow \mathcal{G}(s, a)$
    **return** $r + \gamma \cdot$ ROLLOUT($s', t + 1$)
**end procedure**

---

# E    COMPARISON RESULTS OF SOFT AND HARD CONSTRAINTS

| | Algorithms | KG-A2C-Soft | KG-A2C-Hard | PUCT-RL | MC-LAVE-RL |
|---|---|---|---|---|---|
| *Require-ments* | *Valid Action* | Soft | Hard | Hard | Hard |
| | *Resettable* | - | - | ✓ | ✓ |
| | *External Info* | - | - | - | Word Embedding |
| *Difficult Games* | ZORK1 | $34 \pm 2.2$ | $40.2 \pm 0.4$ | $38.2 \pm 0.8$ | $\mathbf{45.2 \pm 1.2}$ |
| | DEEPHOME | $1 \pm 0.0$ | $20 \pm 2.1$ | $28.6 \pm 2.9$ | $\mathbf{35 \pm 0.6}$ |
| | LUDICORP | $18.6 \pm 0.5$ | $19.8 \pm 1.0$ | $18 \pm 0.0$ | $\mathbf{22.8 \pm 0.2}$ |
| *Possible Games* | PENTARI | $41 \pm 0.9$ | $44 \pm 0.9$ | $64 \pm 2.4$ | $\mathbf{68 \pm 2.0}$ |
| | DETECTIVE | $318 \pm 5.2$ | $\mathbf{338 \pm 3.4}$ | $322 \pm 2.0$ | $330 \pm 0.0$ |
| | LIBRARY | $15.8 \pm 0.5$ | $17 \pm 0.0$ | $\mathbf{19 \pm 0.0}$ | $\mathbf{19 \pm 0.0}$ |
| | BALANCES | $\mathbf{10 \pm 0.0}$ | $\mathbf{10 \pm 0.0}$ | $\mathbf{10 \pm 0.0}$ | $\mathbf{10 \pm 0.0}$ |
| | TEMPLE | $\mathbf{8 \pm 0.0}$ | $\mathbf{8 \pm 0.0}$ | $\mathbf{8 \pm 0.0}$ | $\mathbf{8 \pm 0.0}$ |
| | ZTUU | $5 \pm 0.0$ | $5 \pm 0.0$ | $5 \pm 0.0$ | $\mathbf{7 \pm 2.7}$ |

Table 5: Experimental results comparing soft and hard constraints of valid action handicap. KG-A2C (Soft/Hard) indicates the use of valid action handicap as a (soft constraint via entropy loss over the valid actions/hard constraint for the effective action space). All results indicate averages and standard errors over 5 independent runs.

The original KG-A2C (published state-of-the-art, denoted as KG-A2C-Soft in Table 5) uses the valid action handicap as a soft constraint via entropy loss over valid actions, while our MC-LAVE-RL exploits the valid action handicap as a hard constraint that constrains the effective action space directly. For an equivalent comparison in terms of how a valid action handicap is handled, we also implemented KG-A2C-Hard, which uses the valid action handicap as a hard constraint for the effective action space. Table 5 shows that MC-LAVE-RL still significantly outperforms or matches both KG-A2C-Soft and KG-A2C-Hard on the 8 out of 9 games. This result implies that reducing an effective search space alone may not be sufficient, and language-grounded planning ability is the key to solve IF games successfully. Still, care must be taken to interpret the result of Table 5 in that KG-A2C is a pure RL method and MC-LAVE-RL is an RL algorithm that requires an additional planning assumption (i.e. resettable simulator).

## F  ILLUSTRATIVE BEHAVIOR EXAMPLES

To understand how MC-LAVE-RL discovers more novel states than other reinforcement learning agents, we provide illustrative examples that represent the behavior of each agent playing ZORK1.

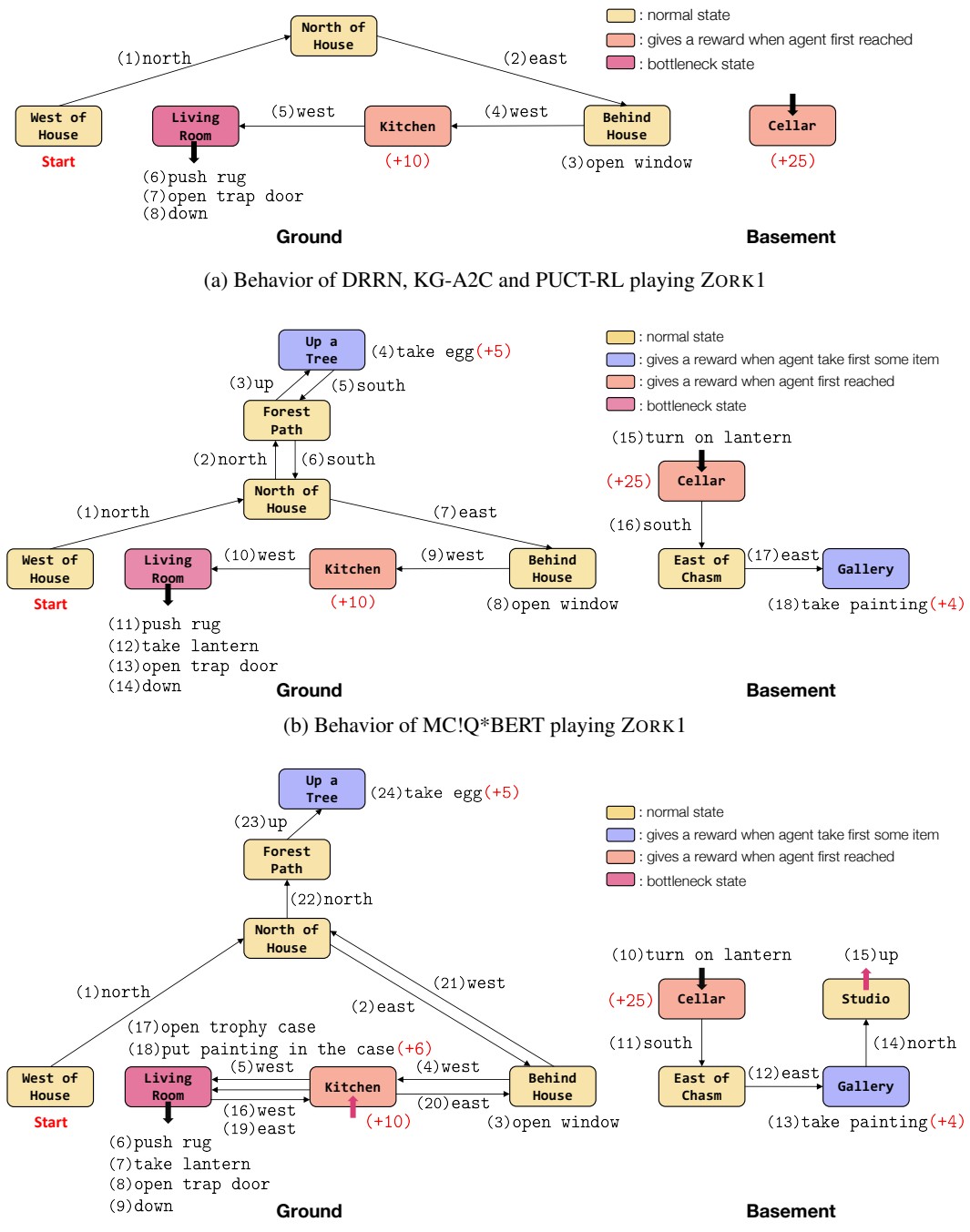

Figure 6: Illustrative examples on ZORK1 that represents the behavior of learned policy by baselines and our algorithms.

# G    ILLUSTRATIVE EXAMPLES FOR LANGUAGE-ACTION EMBEDDING

We present the t-SNE visualization for the language-action embedding and the (semantic) neighborhood of the action 'take lantern' on ZORK1. In the following figures, the color of points indicates the observation-marginalized Q-value of each action with respect to the replay buffer, which is defined as:

$$\mathbb{E}_{(\bar{o},\bar{a})\sim\mathcal{D}(a)}[Q(\bar{o},\bar{a})] \text{ s.t. } \mathcal{D}(a) = \{(\bar{o},\bar{a}) : a = \bar{a} \text{ where } (\bar{o},\bar{a},\bar{r},\bar{o}') \in \mathcal{D}\}$$

Only a subset of the entire language actions is presented in the figures to prevent the cluttered presentation. The bold fonts with cyan and yellow indicate the (semantic) neighborhood of the action 'take lantern' and 'open trap door', and the bold font with magenta indicates the common neighborhood of them. As shown in Figure 7, the semantic neighborhood of language actions defined by the pre-trained word embedding is composed of semantically similar actions. However, as shown in Figure 7, when using the word embedding fine-tuned by $Q_\phi$ training, the semantic neighborhood includes a number of irrelevant actions. This result highlights the importance of word embedding for effective information sharing in MC-LAVE-RL.

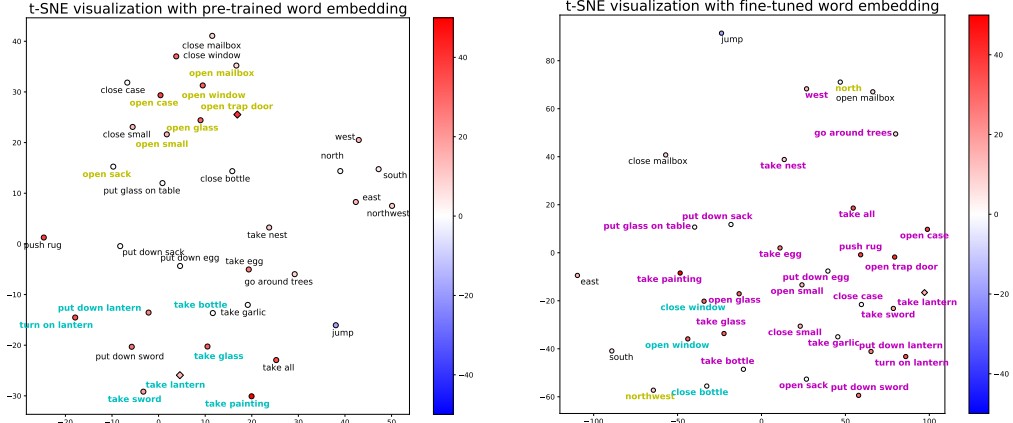

Figure 7: (Left) The t-SNE visualization for the semantic neighborhood of language actions defined by the pre-trained word embedding. The language-actions sharing similar meanings are located at similar points in the embedding space. (Right) The t-SNE visualization for the neighborhood of language actions defined by the word embedding fine-tuned by $Q_\phi$ training. The training objective of $Q_\phi$ does not account for the semantics of language actions but only considers rewards, thus the resulting embedding loses the linguistic semantics of words.

# H    ILLUSTRATIVE EXAMPLE FOR SEMANTIC-BASED INFORMATION SHARING

| MC-LAVE | east | open case | open trap door | take all | take lantern | take sword | turn on lantern |
|---|---|---|---|---|---|---|---|
| $Q(h,a)$ | 7.86 | 10.59 | 14.10 | 2.30 | **14.23** | -1.33 | 13.11 |
| $\pi_\theta(a\lvert o)$ | 0.08 | 0.08 | 0.50 | 0.08 | 0.08 | 0.08 | 0.08 |
| $N(h,a)$ | 23 | 58 | 85 | 47 | 67 | 7 | 63 |
| $L_\phi(a)$ | 13.21 | 35.55 | 25.35 | 38.21 | 49.81 | 2.25 | 34.72 |

| MC-LAVE | east | open case | open trap door | take all | **get lantern** | take sword | turn on lantern |
|---|---|---|---|---|---|---|---|
| $Q(h,a)$ | 6.22 | 11.37 | 14.29 | 4.05 | **15.31** | -0.82 | 13.88 |
| $\pi_\theta(a\lvert o)$ | 0.08 | 0.08 | 0.50 | 0.08 | 0.08 | 0.08 | 0.08 |
| $N(h,a)$ | 21 | 47 | 125 | 35 | 57 | 11 | 54 |
| $L_\phi(a)$ | 13.21 | 35.55 | 25.35 | 38.21 | 41.43 | 2.25 | 34.72 |

Table 6: An illustrative example of search results of MC-LAVE on bottleneck state in ZORK1 (see Figure 1) with modifying the action 'take lantern' to 'get lantern'. The above result with the action 'take lantern' is the same as in Table 3. $Q(h,a)$ denotes the average of Monte-Carlo returns, $\pi_\theta(a\lvert o)$ represents policy prior, $N(h,a)$ represents visit count of each action, $L_\phi(a)$ represents language-driven exploration bonus of MC-LAVE. Note that some differences in $Q(h,a)$ and $N(h,a)$ between results are due to two independent simulation runs.

In order to show the effectiveness of language-action embedding, we further investigate MC-LAVE planning on the bottleneck state presented in Figure 1 and Table 3. Specifically, in the bottleneck state, we artificially modified the action 'take lantern' to 'get lantern' within the valid action set which is used for MC-LAVE planning. As shown in Table 6, MC-LAVE still can successfully select the modified 'get lantern' over 'open trap door' as a final action, even though 'get lantern' had never seen by the agent before. This *generalization* property of MC-LAVE is due to the effect of pre-trained embedding, which shows the importance of language-action embedding for information sharing across semantic space.

