# OpenReview forum: "Monte-Carlo Planning and Learning with Language Action Value Estimates"
_ICLR.cc/2021/Conference — ICLR 2021 Poster_

### Official Review · AnonReviewer4 · 2020-10-15
**Language Value Estimates in MCTS Search**

**Rating:** 7
**Confidence:** 4

**Review:**

This paper presents a method for combining planning an learning in text-based games. In particular it augments Monte-Carlo Tree Search to include a language-similarity bonus to encourage exploration of similar actions. This bonus works by computing a Language Action Value Estimate - which is based on increasing the score of an action by an amount corresponding to the Q-Values of similar actions the agent has experienced. Similarity here is defined by cosine distance in action-embedding space. Using this augmented MCTS algorithm the authors introduce their MC-LAVE agent which alternates between MCTS planning and policy training via supervised learning from the planned trajectories. Experiments across nine IF games show consistent improvement relative to prior RL and planning-based agents. Additional analysis is performed to show how MC-LAVE uses the language action value estimates to learn how to overcome a notable bottleneck in the game of Zork.

I found this paper is well-motivated and principled in its application of Monte-Carlo methods to text-based games. The results show clear improvement over previous agents on a large majority of games and the utility of the Language Value Estimator is clearly shown by the comparison against PUCT-RL.

I have read the author rebuttal and appreciate the changes made to clarify the distinctions and handicaps used by each of the algorithms. Additionally the exploration into ranges of the delta parameter was appreciated.

My largest issue with the paper is that it does not make a clear distinction between pure-RL methods (DRRN/TDQN/KG-A2C) and methods that leverage additional functionality/handicaps to make planning possible (MC!Q*Bert/PUCT-RL/MC-LAVE-RL). Reinforcement learning and planning are different paradigms and even though MC-LAVE-RL does both planning and policy learning - its use of the MCTS planner makes it not an apples-to-apples comparison with the pure RL methods. To this end, I'd strongly encourage the authors to include a discussion of which Jericho handicaps are used by MC-LAVE and make a clear distinction in the presented results between the planning and the pure-RL agents.

The delta hyperparameter for neighborhood size is not given in Table 4 - it would be good to understand what value was used in practice and how large was the effective neighborhood. For example, it would be interesting to understand the effective edit distance in terms of the number of words that could be substituted while remaining in the same neighborhood - e.g. are "take lantern" and "take egg" in the same neighborhood? Taking this idea further - I would be interested to understand how algorithmic performance changes as a function of the neighborhood size - I imagine there may be a sweet spot in terms of this hyperparameter.

---

> ### Author Response · Authors · 2020-11-21
> **Response to R4**
>
> Thank you for your detailed review and constructive feedback. In case you have additional questions and concerns to our response below, we are happy to provide additional response during the rebuttal period.
>
> [Distinction between pure-RL methods]
>
> The only additional assumption we made to make MC-LAVE planning possible is a resettable simulator and a pre-trained word embedding. We added clarifications of the handicaps required for each algorithm in Table 1.
> While reinforcement learning and planning are considered different paradigms, they have commonly been compared on the same line (e.g. [1,2,3]), especially when the model-based simulation is available and the focus is on the eventual performance of the algorithms due to task difficulty. For example, in the game of Go, an RL algorithm RLGO [2] had made a direct comparison to the traditional search-based algorithms. Also in [3], planning-based approaches and pure RL approaches were compared directly in the challenging physical construction tasks. In a similar context, we consider IF games to be challenging yet “simulatable” tasks, and compare our planning-based RL agent MC-LAVE-RL to other RL algorithms in terms of asymptotic performance.
> Furthermore, as presented in Table 1, MC-LAVE-RL mostly outperforms or matches the strong baseline MC!Q*BERT, which makes a similar assumption about the simulator and exploits the pre-trained ALBERT with an external corpus.
>
> [Delta hyperparameter]
>
> For all experiments, we used $\delta=0.3$, which is now included in Table 4 of Appendix B. We also conducted additional experiments to show the effect of varying $\delta$ in Zork1 (Section 5.4, Figure 3), where too small or too large $\delta$ are not effective. This is due to the fact that (1) when $\delta = 0$, no sample in the replay buffer is considered as neighborhood, yielding LAVE exploration bonus to 0, (2) when $\delta = 2$ (the largest possible value), every sample in the replay buffer is considered as neighborhood, yielding LAVE exploration bonus $L(a)$ to a constant value for any $a$. As a consequence, with either two extremes of $\delta$, the MC-LAVE action selection rule of Eq (6) is reduced to the PUCT action selection rule of Eq (2). Finally, Figure 3 presents that there is a sweet spot between the two extremes, which highlights the importance of information sharing between meaningfully defined semantic neighborhoods to solve IF games efficiently.
>
> Furthermore, to better understand the effect semantic neighborhood, we added an illustrative example (Appendix G, Figure 7) that represents the embedding space of actions. Figure 7 shows the embedding space for some of the valid actions including actions in the bottleneck state, and the bold indicates those whose distance from “take lantern” is less than 0.3 (i.e. semantic neighborhood of “take lantern”). As shown in Figure 7, the neighborhood of “take lantern” includes semantically similar actions to “take lantern”, and the language-driven exploration is affected by actions with high value such as “take painting”.
>
> [1] Anthony et al., Thinking Fast and Slow with Deep Learning and Tree Search, NIPS 2017
>
> [2] Silver et al., Temporal-difference search in computer Go, Machine Learning, 2012
>
> [3] Bapst* and Sanchez-Gonzalez* et al., Structured agents for physical construction, ICML 2019

---

### Official Review · AnonReviewer3 · 2020-10-17
**Small modification with interesting results**

**Rating:** 6
**Confidence:** 4

**Review:**

This paper introduces Monte-Carlo planning with language action value estimates to guide exploration. The method builds on top of MCTS w/ PUCT, where a policy distribution over actions is introduced to estimate Q for actions not seen during sampling. The modification proposed here is an additional term to the Q estimate, which is a weighted average of Q values of similar actions, where similarity is computed using word embeddings of words that make up an action. The authors show gains over MCTS  w/ PUCT on 8/9 real text games.

The modification is small, but it is intuitive and shows consistent gains over the baseline without this modification. Some concerns I have are:

1. what is the variance for the experiments in Table 1?
2. sample size of 3 is very small, can you increase this and report the mean/variance?
3. it seems like a single bottle-neck state is crucial for achieving good Zork1 performance. Is this the case with the other two "difficult games" DeepHome and Ludicorp? Do the authors have qualitative observations as to why this method helps on these two games?
4. this paper is missing what I think is a very relevant citation in Branavan 2012 (https://arxiv.org/abs/1401.5390), which uses language from a game manual to guide MCTS.

---

> ### Author Response · Authors · 2020-11-21
> **Response to R3**
>
> Thank you for your detailed review and constructive feedback. In case you have additional questions and concerns to our response below, we are happy to provide additional response during the rebuttal period.
>
> (1 and 2) We ran two more independent trials for each domain so that each result is averaged over 5 runs, which are in Tables 1 and 2. While previous works [1,2,3] have never presented the variation of their results (e.g. standard deviation or standard error), we present the standard error of our results in the revised manuscript (Appendix E, Table 5).
>
> (3) Deephome and Ludicorp also have bottleneck states, and overcoming such bottleneck states is crucial for achieving good performance. Deephome has a bottleneck state related to a lantern, similar to Zork1: Taking a lantern (or torch) does not yield an immediate reward, but the light source plays an important role in the long term to achieve high rewards for navigating in the house after getting into the door. Ludicorp has a similar bottleneck state: the action 'take fuse' does not yield an immediate reward, but it is used in the distant future, where plugging the fuse into the diesel generator gives a large reward.
>
> (4) Thank you for suggesting the relevant reference. We added a short discussion about the work in Section 6. This work has a similar purpose to our work in that it learns an action-value function leveraging the prior knowledge and uses it for Monte-Carlo search. However, unlike MC-LAVE which uses general information of natural language, they assume a text document for high level advice on the task, such as a game manual.
>
> [1] Ammanabrolu and Hausknecht, Graph Constrained Reinforcement Learning for Natural Language Action Spaces, ICLR 2020
>
> [2] Hausknecht et al, Interactive Fiction Games: A Colossal Adventure, AAAI 2020
>
> [3] Ammanabrolu et al, How to Avoid Being Eaten by a Grue: Structured Exploration Strategies for Textual Worlds, https://openreview.net/forum?id=eYgI3cTPTq9

---

> ### Comment · AnonReviewer3 · 2020-11-22
> **Response acknowledged.**
>
> I acknowledge I've read the authors' rebuttal. I maintain my original score.

---

### Official Review · AnonReviewer1 · 2020-10-28
**Review 1**

**Rating:** 4
**Confidence:** 4

**Review:**

Summary: This paper presents a method of incorporating prior knowledge into MCTS via language, using interactive fiction games as a test bed. Their method MC-LAVE used word embeddings on the language action space to help induce a non-uniform distribution over the action space.

Pros:
1. The motivation is very clear, MCTS is generally action agnostic and using language to provide additional semantic information to it can prove to be very effective. The search + RL paradigm has already been shown to work well in cases like Go. The idea of using cosine similarity in word embeddings is a simple but effective way of biasing the MCTS in the right directions.
2. The paper in general is well-written and easy to follow, the qualitative analysis and the additional diagrams in the appendix illustrating the variations in policies are appreciated.

Cons:
Some of the claims are not quite accurate even when compared to the works already cited here -
1. PUCT-RL is the only directly comparable baseline given action space and other handicap differences.
(i) It appears that MC-LAVE is using the valid action handicap in Jericho as a *hard constraint* (Eq. 6 and Algorithm 1) - this means that the MC-LAVE only has a search space of on average < 100 actions per step. The other baselines all use the full template-based action space (except the DRRN) of size 10^8 - a auxiliary entropy loss is used there derived from the valid actions but it is not a hard constraint. As noted in contemporary works such assumptions dramatically reduce the difficulty and language understanding capabilities of text games (Yao et al. 2020 https://arxiv.org/abs/2010.02903).
(ii) The second issue is that MC-LAVE assumes that the simulator is deterministic and can conduct rollouts and reset within the span of an episode - standard planning assumptions but incompatible with all other baselines (except for MC!Q\*BERT) which do not use this handicap.
(iii) Some suggested baselines that make these assumptions would be a heuristic A\* search, or modifying any of the existing algorithms to use smaller action spaces and/or apply alternative exploration strategies seen in previous works such as modular policy chaining (that MC!Q*BERT uses) or Go-Explore (Madotto et al. https://arxiv.org/abs/2001.08868)
2. Even with the results that already exist, it is claimed (for example in section 5.1) that MC-LAVE-RL is the only algorithm to pass bottlenecks such as getting the action "take lantern" right. But the diagrams in the appendix for the policy the MC!Q*BERT agent learns as well as the original paper for that agent show otherwise?
3. Other concerns along these lines: all of this paper's results are averaged over 3 runs while the other baselines are over 5 runs - an indication of variance would be useful to assess whether the differences are significant, especially since some of the margins quite small (23 on MC-LAVE-RL vs 22.8 for the next best on Ludicorp) added to the fact that hyperparameters are different for each game - does that imply that the authors tuned the hyperparameters for each game? The analysis shows that it outperforms only all the other baselines only on 5/9 games and matches on 3. The abstract and intro claim state of the art across all games. I think this means that some amount of claim rewriting is required in addition to the changed baselines.

Overall, I think this paper has a clear motivation and some interesting ideas on how to incorporate semantic language information into planning algorithms. However, in its current state - the comparisons made are not meaningful which makes the claim of state of the art tenuous (state of the art does not matter so much as showing that you make progress in line with the motivation). Making these comparisons would require a heavy rewrite starting from the abstract to the analysis and so I would recommend reject right now but look forward to seeing an updated version of the paper in the future with some of these changes.

After author response:
See reply comment in the thread below for further score justification.

---

> ### Author Response · Authors · 2020-11-21
> **Response to R1**
>
> Thank you for your detailed review and constructive feedback. In case you have additional questions and concerns to our response below, we are happy to provide additional response during the rebuttal period.
>
>  1.
>  (i) We would like to emphasize that all the algorithms in Table 1 are leveraging the 'valid action handicap' provided by the Jericho environment, with the only difference in how they are incorporated into the algorithm, i.e. encoded as a soft constraint using the entropy loss, or as a hard constraint defined as admissible actions. In other words, all the algorithms are using the exact same amount of the information. The exact way this information being leveraged is in fact a design choice dependent on the underlying RL algorithm (If hard constraint works better, why insist soft constraint?). Thus we strongly believe that we can still compare all the algorithms in Table 1 in one table.
>
>  Yet, in order to address the reviewer’s concern, we re-implemented KG-A2C (formally published state-of-the-art at this moment) so that the valid-actions are used as hard constraints rather than soft constraints. We observed that MC-LAVE-RL still significantly outperforms or matches the KG-A2C with using hard constraint on 8 out of 9 games. This new result is added to (Appendix E, Table 5) in the revised paper.
>
>  (ii) MC-LAVE does NOT assume a deterministic simulator. The only assumption we made to make planning possible is a resettable simulator, which is weaker than assuming a deterministic simulator (in the sense that if the dynamics are deterministic, the agent can always backtrack/reset to previously visited states by replaying a prefix of action sequence from the initial state). Still, we acknowledge that MC-LAVE has an additional requirement compared to baseline model-free RL algorithms, except MC!Q*BERT which assumes deterministic simulator, external corpus, and pretrained language model. More clarifications regarding the handicaps required for each algorithm are added to the revised paper (Table 1).
>
>  (iii) Thank you for your suggestions for additional baselines. We conducted additional experiments on KG-A2C which uses a valid-action handicap as a hard constraint. As can be seen from (Appendix E, Table 5), MC-LAVE-RL still significantly outperforms or matches this new baseline on 8 out of 9 games.
>
> 2. You are right. MC!Q*BERT agent, which is concurrent with our work, is also capable of overcoming the bottleneck state as shown in Appendix E. We revised our manuscript to clarify this in Section 5.1 and Section 6.
>
> 3. We ran two more independent trials for each domain so that each result is averaged over 5 runs to match the reported performances of other baseline algorithms, which are in Tables 1 and 2. While previous works [1,2,3] have never presented the variation of their results (e.g. standard deviation or standard error), we present the standard error of our results in the revised manuscript (Appendix E, Table 5). As for the hyperparameter selection, we basically adopted the hyperparameter tuned for Zork1 for all domains, and slight modification is made for each domain to handle the domain-specific settings, e.g. the reward scale in Detective is quite large, thus c_puct should also have become large accordingly. As for the state-of-the-art claim, our algorithm MC-LAVE-RL outperforms or matches the state-of-the-art in various domains except for Ztuu, while requiring an additional assumption of a resettable simulator (please additionally refer to our response to R4 on this matter). We clarified the handicaps each algorithm requires in the manuscript (Table 1).
>
>  [1] Ammanabrolu and Hausknecht, Graph Constrained Reinforcement Learning for Natural Language Action Spaces, ICLR 2020
>
>  [2] Hausknecht et al, Interactive Fiction Games: A Colossal Adventure, AAAI 2020
>
>  [3] Ammanabrolu et al, How to Avoid Being Eaten by a Grue: Structured Exploration Strategies for Textual Worlds, https://openreview.net/forum?id=eYgI3cTPTq9

---

> > ### Comment · AnonReviewer1 · 2020-11-23
> > **R1 Rebuttal Response**
> >
> > Thank you for the work you have done in running the additional experiments and revising the manuscript.
> >
> > Two major concerns, and why I do not think a direct comparison across all things in Table 1 is warranted still:
> > 1. This is to do with point 1. (ii) of my initial review. The bigger handicap here is the fact that planning/search algorithms are being compared to RL algorithms. These have fundamentally different assumptions, assuming that the simulator is resettable at will effectively to anything but the starting state is different problem than what the RL agents are trying to solve. There's nothing wrong with making the kinds of assumptions that the authors have made (deterministic simulator/hard constraint/etc) but comparing it to those that do not in a single table is a bit of an apples to oranges comparison.
> > 2. The KG-A2C with hard constraint is not a very good baseline given that it was never designed to exploit the hard constraint in the first place. It was designed to account for the additional difficulty in having the soft constraint by filtering actions via the graph - once this is done by the hard constraint there is little separating KG-A2C from vanilla A2C (the original paper indicated that the KG input representation portion did not account for any increase in final score) and in fact some games received an increase in score when the KG/soft constraint combination was relaxed. Once again, there is nothing inherently wrong with any of these methods - just that it becomes an issue when you attempt to compare to it side by side.
> >
> > This being said, the handicap clarifications added in table 1 do help, though the prose still compares them all to each other directly. PUCT-RL and MC-LAVE-RL are the only two directly comparable things there. In appreciation of the authors efforts during the rebuttal phase, I will raise my score though overall still do not think this paper is ready for acceptance.

---

> > > ### Author Response · Authors · 2020-11-24
> > > **Response to R1**
> > >
> > > Thank you for the additional feedback. In addressing your major concerns:
> > >
> > > - Regarding the comparison made in Table 1, we agree with the reviewer that different algorithms work under different assumptions. There are currently only a handful of algorithms on IF games, but we believe that we will be able to make comparison to many directly relevant algorithms as the domain attracts more attention in the RL/planning community. We hope that the “requirements” row in table 1 and the related discussion in page 7 (highlighted in red) in the revision helps readers understand the advantages and limitations behind these approaches.
> > >
> > > - Given that all algorithms are *equally* using the valid action handicap (via either soft or hard constraint), please note that we also make comparison to a strong baseline MC!Q*BERT (concurrent work with ours) in the main text, which requires a similar assumption on a resettable simulator as ours and stronger requirements on the external knowledge such as pre-trained ALBERT and Jericho-QA corpus. We have incorporated the comment related to the pros and cons of KG-A2C + hard constraint in Appendix E, where we show the result.

---

### Official Review · AnonReviewer5 · 2020-11-09
**Neat idea with promising results, but deserves deeper experimental investigation**

**Rating:** 7
**Confidence:** 4

**Review:**

This paper presents a novel MCTS-based policy improvement operator called MC-LAVE designed specifically for environments with text-based action spaces. MC-LAVE adds an additional term to PUCT that shares information across semantically similar actions. This additional term for a given action $a$ is set to the soft maximum over the Q function evaluated at all $(o, \bar{a})$ pairs of transitions in the replay buffers whose transition action $\bar{a}$ is within some cosine distance $\delta$ of the action $a$.  As shown in Appendix C, addition of this term preserves the expected regret bounds of PUCB. The paper reports empirical improvements over existing methods on various interactive fiction (IF) games in the Jericho suite.

The results of MC-LAVE are promising, however, the paper leaves me with a few lingering questions that I feel should be resolved before I would feel this paper were ready for conference publication:

1. It is not clear to me that the improvements are, as repeatedly emphasized by the authors throughout the paper, due specifically to MC-LAVE leveraging semantic similarity of useful actions to focus on the most promising actions: It is possible that the additional term is simply injecting additional randomness to the action selection rule, which benefits exploration. To control for this possibility, the authors should include a baseline that performs the action selection rule of MC-LAVE with the semantic neighborhood N(a) of action $a$ set to a uniformly randomly sampled set of valid actions in each corresponding state of the replay buffer.

2. Related to the above point, the hyperparameter $\delta$ seems to be quite important, as it effectively defines the size of the neighborhood. However, neither the exact setting of $\delta$ used in the experiments nor the effect of varying $\delta$ is presented in the paper. Relatedly, another important baseline similar to the random neighborhood baseline mentioned above is setting $\delta$ to a very large number.

3. I am not entirely convinced that the action-spaces of the IF games present a suitable testbed for the semantics-based information sharing that MC-LAVE attempts to achieve, as the action space vocabulary for valid actions in each state seems quite limited and tending to repeat the same key verbs, based on the presented examples in Table 3. Perhaps some comment about the diversity of these games could be made to defend against this criticism, or perhaps the action space of the games could be edited to exhibit a more diverse vocabulary to allow better empirical demonstration of the effectiveness of MC-LAVE in performing semantics-based info sharing across related actions.

4. How important are the embeddings? Based on the presented examples, it is not clear to me how the semantics-based info sharing defined in the MC-LAVE action selection rule should benefit the agent in choosing "take lantern" over "open trap door" in a meaningful way beyond just recognizing that "take" is often useful in other, albeit, completely different contexts. In that case, perhaps word embeddings are not even necessary, but rather maintaining a table of Q-values for $(o, a_i)$ for each action word token $a_i$ and taking the soft maximum over the corresponding Q-values of action tokens might suffice.

5. I would love to see some examples of how the language-driven exploration term in the MC-LAVE action-selection rule weighs the various action tokens in a fully-trained policy, and which words from other states in the replay buffer the words are being linked to. I think such a qualitative analysis would be useful in better understanding how the "language-driven exploration" term enables the agent to attain the empirical gains reported.

6. It's not clear how MC-LAVE, even if effectively sharing info between semantically-related words at each node of the MCTS tree, is improving "non-myopic exploration" over alternative methods, which point the authors seem to emphasize throughout the preliminary sections of the paper.

7. Some discussion around why the authors think MC-LAVE underperforms w.r.t. existing methods on ZTUU would improve the discussion of the experimental results.

---

> ### Author Response · Authors · 2020-11-21
> **Response to R5**
>
> Thank you for your detailed review and constructive feedback. In case you have additional questions and concerns to our response below, we are happy to provide additional response during the rebuttal period.
>
> 1. Leveraging the semantic similarity of useful actions is particularly important for MC-LAVE to solve IF games. To see that MC-LAVE leverages semantic similarity rather than injecting random noise, we conducted additional experiments on the baseline proposed by the reviewer, which was made to draw 10 samples uniform randomly from the replay buffer to form the set of neighborhoods $N(a)$ of action $a$. We observed that the performance of the proposed baseline was almost identical to PUCT, which significantly underperforms MC-LAVE. This is because the set of randomly sampled neighborhoods works as useless random noise, failing to provide meaningful information to improve search efficiency.
>
> 2. For all experiments, we used $\delta=0.3$, which was tuned for Zork1, and this same hyperparameter was used for all other domains (now included in Appendix B). We also conducted additional experiments to show the effect of varying $\delta$ in Zork1 (Section 5.4), where too small or too large $\delta$ are not effective. This is due to the fact that (1) when $\delta = 0$, no sample in the replay buffer is considered as neighborhood, yielding LAVE exploration bonus to 0, (2) when $\delta = 2$ (the largest possible value), every sample in the replay buffer is considered as neighborhood, yielding LAVE exploration bonus $L(a)$ to a constant value for any $a$. As a consequence, with either two extremes of $\delta$, the MC-LAVE action selection rule of Eq (6) is reduced to the PUCT action selection rule of Eq (2). Finally, Figure 3 presents that there is a sweet spot between the two extremes, which highlights the importance of information sharing between meaningfully defined semantic neighborhoods to solve IF games efficiently.
>
> 3. The action-spaces of IF games are defined by the environment parser's vocabulary whose size is typically from 500 to 2200. While the size of the vocabulary is limited compared to large-scale open domain dialogues, it still serves as a useful testbed to demonstrate the effectiveness of semantic-based information sharing of MC-LAVE. For example, the Zork1 environment produces identical observations for taking the actions 'put painting in the case' and 'place art into the case', where MC-LAVE can readily take advantage by exploiting the information sharing across those semantically similar actions. Scaling MC-LAVE to more challenging testbeds with real-world sized vocabulary remains as an interesting direction for future work.
>
> 4. The embeddings are particularly important for MC-LAVE. For example, in the referred bottleneck state, we observed that even though we artificially modified the action 'take lantern' to 'get lantern' (these are treated the same by Jericho environment), MC-LAVE still could successfully select the modified 'get lantern' over 'open trap door' as a final action. This generalization property of MC-LAVE could not have been observed without the pre-trained embeddings, which promotes information sharing across semantic space. Besides, maintaining a table of Q-values for each word token $Q(o, a_i)$ is not straightforward since the transitions and rewards of the environment are defined only for the full language action (i.e. a sequence of words), not for a single word token.
>
> 5. Table 3 presents how the language-driven exploration term $L(a)$ of MC-LAVE differs for each language action in the bottleneck state of Zork1. Please note that $L(a)$ is defined for the full language-action (i.e. a sequence of words), not for each word token of the language-action. We also additionally provided a qualitative analysis of how language-actions were linked to each other in the embedding space via t-SNE visualization in Appendix G.
>
> 6. Please note that we did not make a claim that MC-LAVE's information sharing leads to non-myopic exploration; We stated that the epsilon-greedy (or softmax sampling) action-selection strategy of baseline RL algorithms lack structured and non-myopic planning ability. In contrast, the structured and non-myopic exploration of the MC-LAVE-RL agent is **due to conducting Monte-Carlo tree search**, rather than from information sharing. MC-LAVE's information sharing among semantically similar actions allows for further improvement in the search efficiency of MCTS.
>
> 7. We found that MC-LAVE's poor performance on Ztuu is attributed to the limited search budget. To achieve DRRN's reported score, the agent should reach a specific state that can potentially yield infinite rewards because of a bug in the simulator. This bug can only be reached by taking a precise sequence of actions during long timesteps that give no immediate reward. We observed that MC-LAVE could reach that state with x10 more search budget while keeping the other hyperparameters to be the same.

---

> > ### Comment · AnonReviewer5 · 2020-11-21
> > **Response to rebuttal**
> >
> > Thanks for taking the time to consider my comments and look further into each of these points. However, I find that many of the clarifications above were not included in the revision. In particular, could the authors add the ablation described in (1) as well (perhaps to the Appendix)? Further, could the authors add their points addressing (3) and (4) to the paper? I felt these were important supporting points missing in the original paper, which is why I brought them up. Otherwise, this seems like a well-presented, simple extension of UCB-MCTS with good results, so I have bumped up my score.

---

> > > ### Author Response · Authors · 2020-11-24
> > > **Response to R5**
> > >
> > > Thank you for your feedback. We added our clarifications in the author response to the revised manuscript, which are highlighted in blue.

---

### Author Response · Authors · 2020-11-24
**Summary of the revisions**

We thank all the reviewers for their constructive feedback and comments. The improvements and modifications in the revision are summarized as follows:

**[Experiments]**
1. Clarification of the handicap and requirements for the distinction between pure-RL methods and planning-based RL methods to address the concerns of Reviewer 1 and 4 (Section 5.1, Table 1, Table 5).
2. Additional results and analysis for the effect of size of the semantic neighborhood to address the suggestions provided by Reviewer 4 and 5 (Section 5.3, Figure 3).
3. Additional results and analysis for the random noise injected PUCT to address the suggestion provided by Reviewer 5 (Section 5.3, Figure 3).
4. Update the results with additional runs including standard error to address the suggestions provided by Reviewer 1 and 3 (Table 1, 2, 5).
5. Additional experimental results of KG-A2C-Hard, which uses valid action as a hard constraint, to address the concerns of Reviewer 1 (Appendix E).
6. Additional illustrative examples for language-action embedding to address the concern of Reviewer 4 and 5 (Appendix G).
7. An additional illustrative example for semantic-based information sharing to address the suggestion of Reviewer 5 (Appendix H).

**[Writing]**
1. Clarification of effect between MCTS-based planning and language-grounded exploration to address the suggestion of Reviewer 5 (Section 2.2).
2. Additional explanation of how suitable the testbed is used in the paper to address the concern of Reviewer 5 (Section 5, Appendix H).
3. Revised the explanation of the baseline algorithm (MC!Q*BERT) to address the concern of Reviewer 1 (Section 5.1, Section 6).
4. Added a citation and explanation of related work to address the suggestion of Reviewer 3 (Section 6).

---

### Decision · Program_Chairs · 2021-01-07
**Final Decision**

**Decision:**

Accept (Poster)

**Comment:**

I thank the authors for their submission and very active participation in the author response period. The paper is well motivated, clearly written and demonstrates empirical gains. In discussions, R4 and R5 were championing the paper. R1 stated that the paper improved, but insists that claims of state-of-the-art, given the simplifications induced by a deterministic simulator and hard valid action constraint, are not justified without additional baselines to compare to. The authors have toned down state-of-the-art claims in the revised version of the paper. I agree with R4 that the added requirements in Table 1 sufficiently explain the constraints under which MC-LAVE can be applied. Given the strong positive sentiment by R4 and R5, the positive assessment by R3, and the detailed response by the authors, I am recommending acceptance of the paper.